# Organic Matter Structural Composition of Vascular Epiphytic Suspended Soils of South Vietnam

Evgeny Abakumov [1],* and Alen Eskov [2]

1   Department of Applied Ecology, Saint-Petersburg State University, 16 Line of VO 29,
    199178 St. Petersburg, Russia
2   Department of Plant Ecology and Geography, Moscow State University, Leninskie Gory 1,
    119991 Moscow, Russia
*   Correspondence: e_abakumov@mail.ru or e.abakumov@spbu.ru; Tel.: +7-9111969395

**Abstract:** The biosphere organic matter is stored in nature in various forms. Most of it is associated with classical terrestrial organo-mineral soils. The carbon of woody plant biomass is counted separately from soil as carbon of the standing biomass. Nevertheless, humification as a universal process already begins in plant residues before they reach the surface of the classical tropical mineral soil. Moreover, in tropical humid-forest ecosystems, most of the organic matter does not reach the soil surface at all and accumulates in the state of suspended soils. The data obtained in this study characterize, for the first time, the component and structural composition of the organic matter of plant residues of suspended soils, as well as the products of their transformation—humic substances formed in suspended soils. With the use of micro morphological methods, it was shown that humification appears in merged areas of organic remnants. There were statistically significant differences in the elemental composition of humic acids and initial organic material for all elements—C, H, O and N. It has been shown that the aliphatic part dominates (75–93%) in the initial organic materials of suspended soils, whereas the humic substances are characterized by a relatively increased fraction of aromatic fragments (31–42%) in the composition of their molecules, which confirms that humification takes place. Thus, even in the suspended soils, classical humification occurs, and this is not limited by the low content of mineral particles and cations in the suspended soils and the rather acidic reaction of the material. Therefore, the existence of tiering and the formation of the corresponding layers of suspended soils is accompanied by the stabilization and humification of organic matter, which is accompanied by a radical change in its structural and component composition. This process is the "natural biotechnology" of organic matter conservation and stabilization is discussed in article.

**Keywords:** suspended soils; vascular epiphytes; South Vietnam; humification; 13-C NMR spectroscopy

## 1. Introduction

Soil organic matter is the most important storage for carbon on the planet. It plays very important role in climate mitigation and the stabilization of biogeochemical processes in a whole biosphere. There are numerous works dealing with the characterization of organic matter in "normal" soils, formed on mineral parent materials [1,2]. At the same time, soil formation occurs not only under the folic material of a forest floor located on the surface of mineral soil forming substrata but also in epiphytic organic and organo-mineral formations, typical for tropical wet forests [3]. In these communities, soil formation is stratified in a vertical scale. The first is a "classical" soil [4–6] formed as result of interaction of the mineral part with incoming organic matter; these soils are not rich in organic matter due to the high mineralization rate. One can see that in tropical forest there are no continuous forest floors and the existing folic material on the soil surface is sporadic, loose and shallow. The second type of soil is so-called suspended soils. The main difference between suspended soils and classical mineral soils is that suspended soils have no direct connection with

the mineral part of the landscape, including soil parent materials. The organic mineral nutrition of suspended soils is carried out at the expense of atmospheric precipitation and a partial return of biogenic elements. Thus, suspended soils have both the attributes of soils proper and the properties of soil–substrate organogenic bodies. In the most primitive case (in temperate ecosystems), they are formed from the litter stuck in the forks of skeletal branches [7]. In tropical ecosystems, well-structured communities of vascular epiphytes in tree crowns form the basis of suspended soil clusters. With the help of various adaptations (a special type of roots [8], leaves forming funnels, clumps of outgrowths of epiphytes), they are able to massively trap falling organic material, retain water and provide shelter for settling consort organisms—from fungi to ants and vertebrates [9]. Together with the living biomass of epiphyte, suspended soil forms epiphytic material (EM) [10]. The rate of epiphytic organic matter deposition is impressive: the maximum recorded estimate is 44 t ha$^{-1}$ [10]. This also leads to a large-scale accumulation of mineral nutrients in epiphytic matter. One of the few estimates described provides: N = 37.9 $\pm$ 9.0; P = 1.97 $\pm$ 0.47; K = 9.6 $\pm$ 2.3; Ca = 9.6 $\pm$ 2.3; Mg = 2.64 $\pm$ 0.63; Na = 0.25 $\pm$ 0.06 kg ha$^{-1}$ for a total EM biomass of 2261 $\pm$ 537 kg ha$^{-1}$ [11]. Moreover, recent studies have shown a wide diversity of tropical suspended soil metabolomes [12]. It appears that suspended soils have strong buffer properties and are much less dependent on chemical interference from the outside than terrestrial soils [13]. This makes suspended soils an essential functional unit of the tropical ecosystem, which appears to replace the tropical forest's absent ground litter [14].

Organic matter accumulates in suspended forms, with organic residues intensively transformed; there is evidence of humification, which takes place under conditions of very low mineral particles [15]. Due to the low content of mineral particles of atmospheric origin, only the ash components may play an essential role in binding of humification products and stabilization of organic matter. As was mentioned by numerous researchers [16,17], soil organic matter humification degree depends essentially on quality and composition of organic precursors of humification. Although the end products of humification have been investigated previously [9,15], particular interest is given to the investigation of the quality of the organic precursors of humification.

Thus, the aim of this work was to characterize for the first time the structural and component composition of various plant materials in the suspended soils of South Vietnam. The set of objectives was formulated: (1) to analyse the pattern of elemental composition of bulk organic matter of suspended soils, (2) to characterize these matters with use of 13-C solid state NMR and (3) to investigate the composition of the end organic matter transformation products—humic acids—isolated from selected suspended soils.

## 2. Materials and Methods

### 2.1. Regional Setting

Field studies were conducted in southern Vietnam in 2015–2018 in three different habitats (Figure 1). The first habitat was open savanna-like forest on oligotrophic soils (Phu Quoc). Phu Quoc Island, located in the Gulf of Siam, has a subequatorial, humid climate [18]. Sampling plots with a total area of several hectares were located not far from both shores of the island (within 0.5–3 km), around the line between 10°16′46″ N, 103°55′23″ E and 10°24′17″ N, 104°03′02″ E (altitude 0–1.5 m asl) (Table 1). Average tree height was 5−7 m. Trees were widely spaced, the herb layer poorly developed and alternated with large areas of bare soil. Many trees were infected by hemiparasites from the Loranthaceae and Viscaceae families (three species).

The second habitat investigated was lowland forest (Cat Tien). Cat Tien National Park is located in the lowlands of Dong Nai province. It has a subequatorial monsoon climate [18]. 2013). The sample plot occupied approximately 1 hectare around the point 11°26′14″ N and 107°25′26″ E (≈120 m asl). Canopy tree height reached 30–35 m.

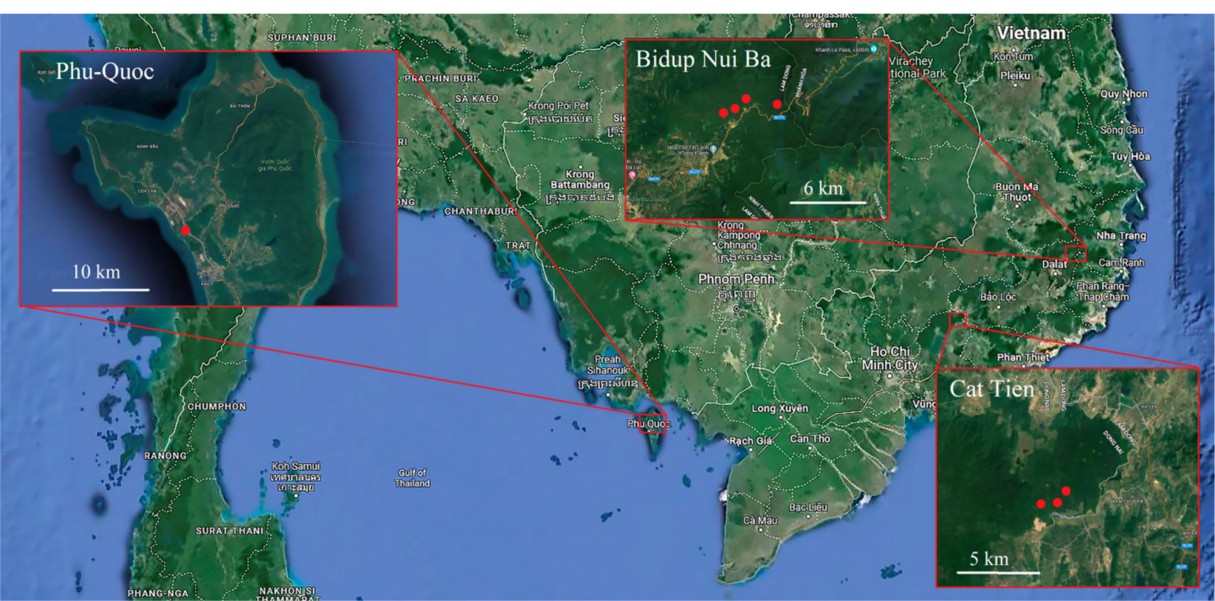

**Figure 1.** Map of the study region. Sample sites are marked with red circles.

**Table 1.** Edaphic and climatic conditions of the habitats studied.

| Location | Phu Quoc | Cat Tien | Bidoup |
|---|---|---|---|
| Mean annual precipitation | 2880 mm | 2470 mm | 1860 mm * |
| Dry season | January–February | November–April | Not pronounced |
| Mean annual temperature | 27.1 °C | 26.2 °C | 18.2 °C |
| Soil | Folic Arenosol, Stagnic | Dystric Skeletic Rhodic Cambisols and Skeletic Grey Umbrisols | Loamic Folic Ferralsols |

* data for Dalat city, somewhat higher in the montane forest.

The third habitat was montane forest (Bidoup). Bidoup Nui Ba National Park is located on Da Lat Plateau. Its climate is much wetter and cooler than in the neighbouring plains of southern Vietnam (Table 1). Materials were collected at Hon Giao mountain top (≈2000 m asl) in a stunted, one-layered cloud forest, from a plot of several hectares around the point 12°11′24″ N and 108°42′38″ E. Average tree height was 5−8 m.

*2.2. Sampling Procedure*

Suspended soil samples were collected in two ways (Figure 2). We picked samples 1-1 through 1-4 from a single tree of *Lyonia ovalifolia* species in the Bidup mountain forest along an altitudinal gradient from the base to the crown under the epiphyte community as a whole without differentiating suspended soils by epiphyte species. Suspended soils up to 2 m (sample 1-1) are formed under predominantly spore plants (mosses and ferns); at 2 to 4 m, they are joined by some flowering epiphytes, primarily orchids (sample 2-2); at 4–6 m, large skeletal branches are located, collected suspended soils are formed in large nesting ferns and large orchids (sample 2-3) and finally at heights of 6–8 branches, we gathered suspended soils in the form of thin crusts under small orchids and lichens. We isolated sample 2-1 under a moss cushion in the Bidup moss forest. Samples were collected using sliding ladders extended to the height of the desired tier and using retractable scissors; after cutting, the samples were caught with a net. We also took the remaining samples (2-2 through 2-7) from a species-specific epiphyte coma of ferns and flowering plants in three habitats. We collected 100–300 g for each sample and put them into paper bags, cleaned them of coarse root and leaf debris and dried them at room temperature over silica gel.

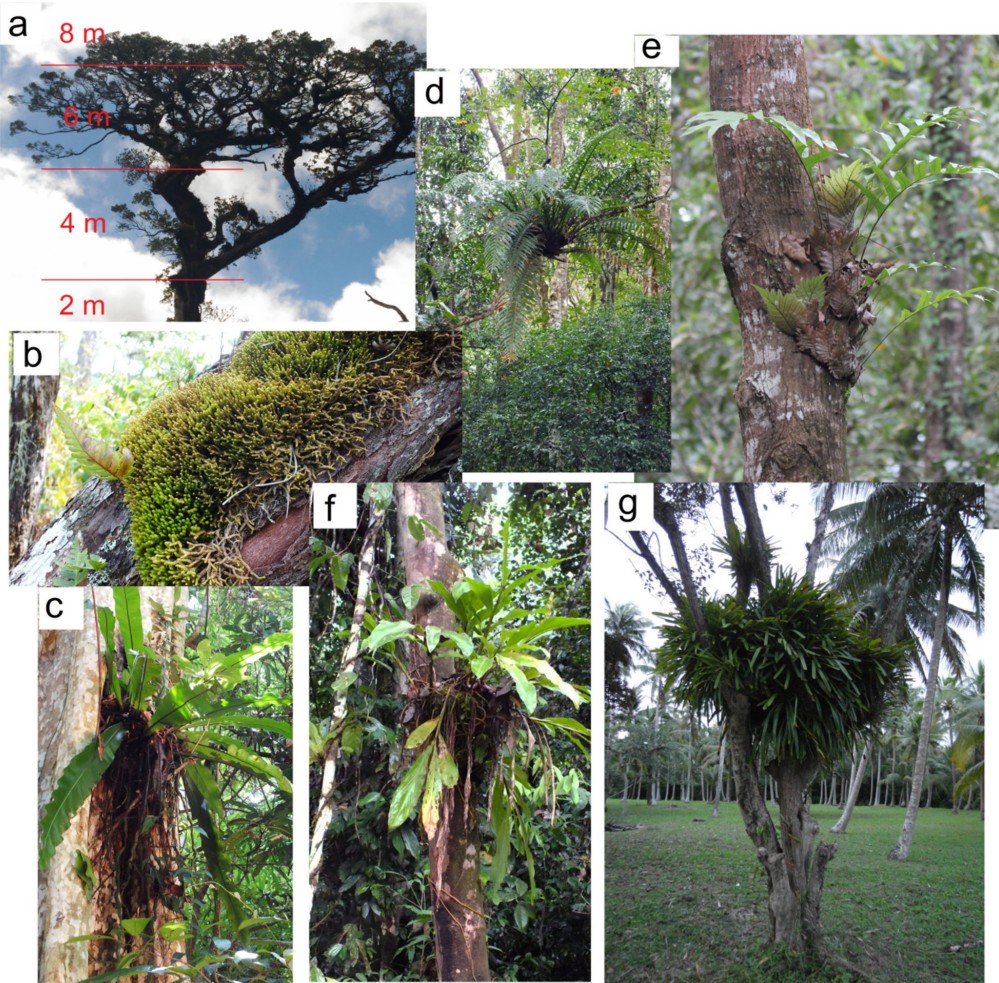

**Figure 2.** Some epiphytic communities and epiphytes from which suspended soils were collected:
(**a**) *Lyonia ovalifolia* tree in np Bidup Nui Ba from which suspended soils were collected from heights
up to 2 m, mainly under mosses and ferns; 2–4 m. moss cushions with ferns and small orchids
underplanted; 4–6 m, mainly large orchids and heather and 6–8 m, mainly small orchids and lichens
on individual thin branches; (**b**) moss cushions with suspended soils form in the mountain forest of
np Bidup Nui Ba; (**c**) classical nesting epiphyte *Asplenium nidus* forms suspended soils in a lowland
forest (Cat Tien park); (**d**) staple epiphyte *Aglaomorpha* sp. in mountain forest of Bidup Nui Ba;
(**e**) staple epiphyte *Drynaria quercifolia* in lowland forest (Cat Tien park); (**f**) nesting epiphyte
*Hedychium bousigonianum* in lowland forest (Cat Tien) and (**g**) nesting epiphyte *Cymbidium finlaysonianum*
in a sparse community (Phu Quoc).

*2.3. Laboratory Analysis*

Solid-state 13C-NMR spectra of the bulk organic matter and humic acids solid powders
were measured with a Bruker Avance 500 NMR spectrometer in a 3,2-mm $ZrO_2$ rotor. The
magic angle spinning speed was 20 kHz in all cases and the nutation frequency for cross
polarization was u1/2p 1/4 62.5 kHz. Repetition delay and the number of scans were
3 s. Humic acids (HAs) were extracted from each sample according to a published IHSS
protocol http://humic-substances.org/isolation-of-ihss-samples/, accessed on 1 March
2022). Briefly, the soil samples were treated with 0.1 M NaOH (soil/solution mass ratio of
1:10) under nitrogen gas. After 24 h of shaking, the alkaline supernatant was separated from
the soil residue by centrifugation at 1516× *g* for 20 min and then acidified to pH 1 with 6 M
HCl to precipitate the HAs. The supernatant, which contained fulvic acids, was separated
from the precipitate by centrifugation at 1516× *g* for 15 min. The HAs were then dissolved
in 0.1 M NaOH and shaken for four hours under nitrogen gas before the suspended solids

were removed by centrifugation. The resulting supernatant was acidified again with 6 M HCl to pH 1 and the HAs were again isolated by centrifugation and demineralized by shaking overnight in 0.1 M HCl/0.3 M HF (soil/solution ratio of 1:1). Next, the samples were repeatedly washed with deionized water until pH 3 was reached and then finally they were freeze-dried. Data were corrected for water and ash content. The elemental composition of HAs is the percentage content of the elements C, H, N and O. For the graphical analysis of the elemental composition, we used the van Krevelen diagram [19], using H/C-O/C ratios to determine the direction of transformation processes of various organic compounds in natural conditions. The elemental composition was corrected for the weight of moisture and ash content. The oxygen content was calculated from the difference in mass of the whole samples and gravimetric concentrations of C, N, H and ash. Determination was performed with use of an elemental analyser (EA3028-HT Euro Vector, Pravia PV, Italy). The oxygen content was determined by the difference in the ratios. Gravimetric concentrations are given for C, H, O and N content. C/N, H/C, O/C and H/C were calculated from the mole fractions of C, H, O and N gravimetric contents. H/C (mod) is the number of substituted hydrogen atoms in HAs; H/C (mod) = H/C + 2(O/C) × 0.67; the W index is the degree of oxidation in molecules [20]. SD ± 0.05 was employed for the contents of C, H and N.

Thin sections (0.03 mm) for micro morphological studies were prepared using the regular grinding wheel and fine (0.8–1.6 microns) abrasive paste from the micromonoliths of soils sampled during several field campaigns. Samples were air-dried and then saturated with epoxy resin. Thin sections were investigated with the use of optical polarization microscopes, Leica DM750P (SPbGU). The terminology used in this paper is published by Gagarina [21] and Gerasimova et al. [22], where details of the micro-organization of soil were described and classified in detail [23].

Data of sample characteristics are provided in Table 2.

**Table 2.** Sample characteristics.

| Sample Code | Short Description | Height of Sampling, m | pH of Grounded Material | Ash, % |
|---|---|---|---|---|
| 1-1 | One tree | 2 | 3.00 | 0.20 |
| 1-2 | One tree | 4 | 3.50 | 0.37 |
| 1-3 | One tree | 6 | 3.20 | 0.45 |
| 1-4 | One tree | 8 | 2.80 | 0.42 |
| 2-1 | Suspended soil of Bidup, mountain massive (Bryophyta) | 4–5 | 3.10 | 0.26 |
| 2-2 | Nest of the fern *Asplenium nidus* (Cat Tien park)—suspended soil of mainly organic composition | 6 | 3.90 | 0.24 |
| 2-3 | Suspended soil of the "staple" fern *Aglaomorpha* (Bidup park) turf (histic) material | 7 | 3.00 | 0.17 |
| 2-4 | Suspended soil of the "staple" fern *Drynaria quercifolia* (Cat Tien), turf (histic) material | 5 | 4.80 | 0,16 |
| 2-5 | Suspended soil of flowering heather (*Vaccinium* sp.) (Bidup). Ericoid mycorrhizae is repeated. Accordingly, these soils are poor acidic turf-like substrates | 10 | 3.40 | 0.53 |

**Table 2.** *Cont.*

| Sample Code | Short Description | Height of Sampling, m | pH of Grounded Material | Ash, % |
|---|---|---|---|---|
| 2-6 | Suspended soil of flowering ginger (*Hedychium bousigonianum*) (Cat Tien). Input of organic matter by insect (ants), this formation is closest to soils | 5 | 3.60 | 0.55 |
| 2-7 | Suspended soil of flowering orchid (*Cymbidium finlaysonianum*) (Phu Quoc) | 2 | 3.80 | 0.52 |

## 3. Results and Discussion

### 3.1. General Characteristics

All the soils investigated were characterized by acid or very acid pH values measured in water suspensions. This could be connected to the low content of ash and base elements in the litter, which is typical for tropical forest [6] compared with oligotrophic terrestrial ecosystems of Podzols in the temperate zone. It should be emphasized that the low content of base cations may affect the low soil organic matter stabilization rate in suspended soils. Data on low pH values are correspond well with the ash values provided in Table 2.

Micromophological organization of the studied samples in thin sections is presented in Figures 3 and 4. These pictures demonstrate that there is some transformation of organic remnants, especially on the merged areas of organic particles. Some of the particles demonstrate diffusive penetration transformation with simultaneous decaying of organic matter. Many of organic particles demonstrate browning of the colour, supposedly dealing with the initial humification of organic matter. Thus, in all studied suspended soils, there is a mechanical destruction of organic material and initial humification that is localized mainly on the edges of plant residues.

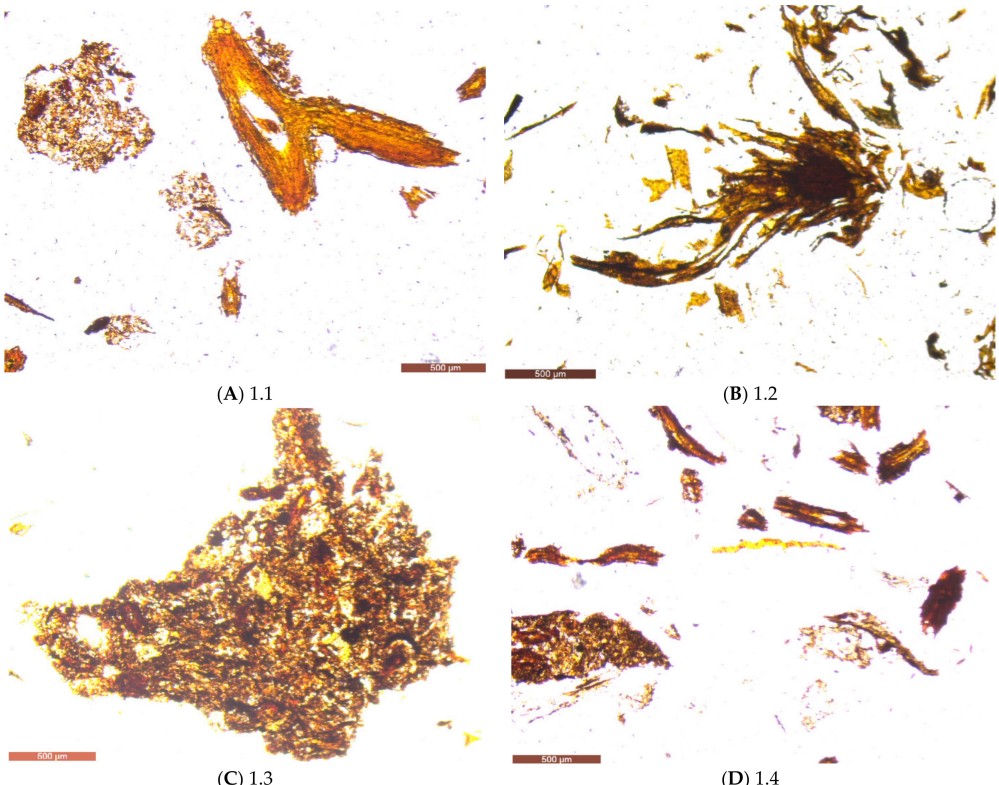

(**A**) 1.1 (**B**) 1.2

(**C**) 1.3 (**D**) 1.4

**Figure 3.** Organic matter morphology. Sample set № 1. Transmitted light. (**A**) 1.1, (**B**) 1.2, (**C**) 1.3, (**D**) 1.4 in Table 2.

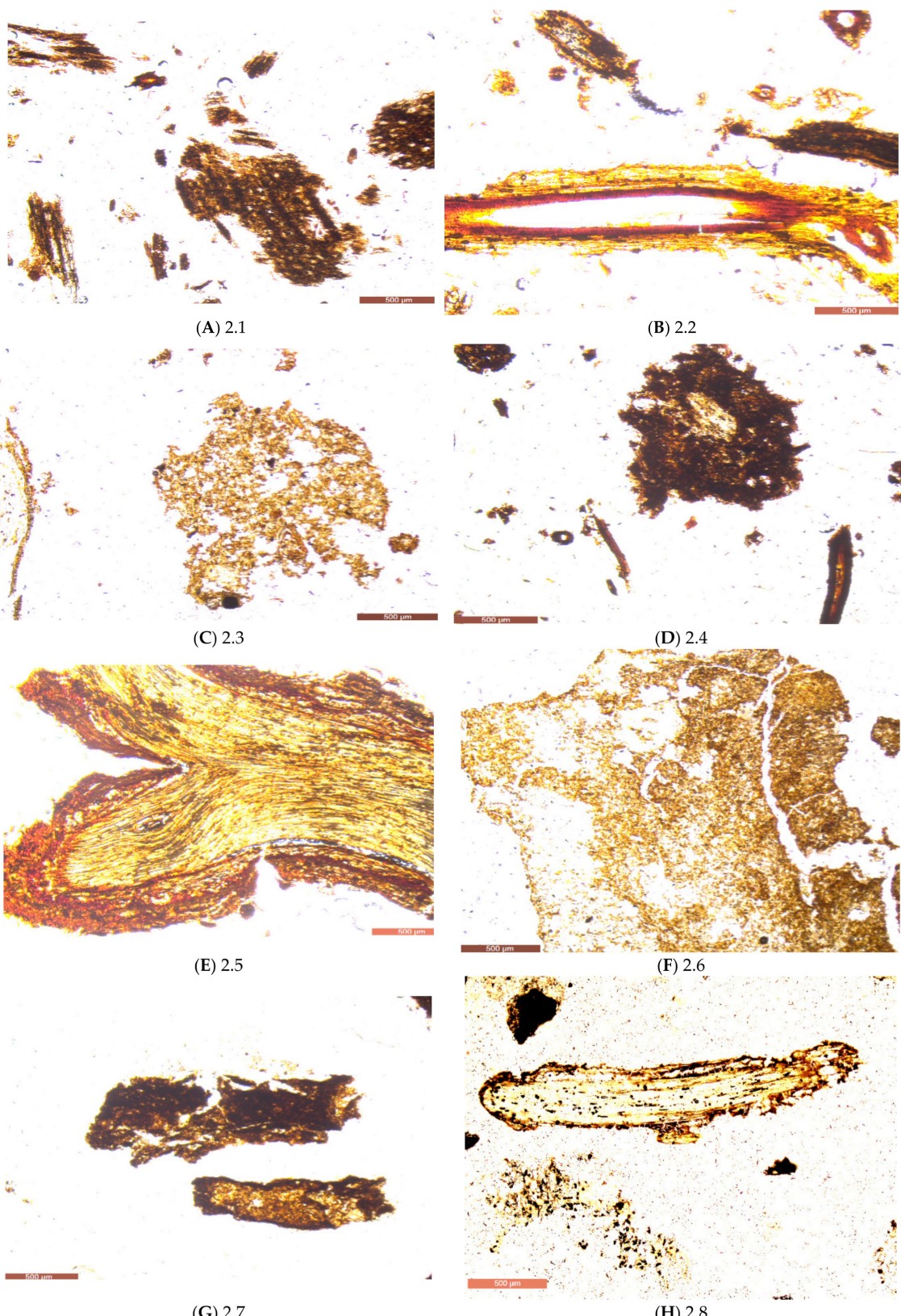

**Figure 4.** Organic matter morphology. Sample set 2. Transmitted light. (**A**) 2.1, (**B**) 2.2, (**C**) 2.3, (**D**) 2.4, (**E**) 2.5, (**F**) 2.6, (**G**) 2.7, (**H**) 2.8.

In other words, suspended soils resemble upper peats to some extent. The question arises: how are they structured into vertical clusters on the tree trunk and able to persis-

tently withstand streams of tropical rains without being subjected to erosion? As some of our studies show, a structural element that may achieve this is the agraviotropic root of epiphytes, which forms the framework for holding the epiphytic clump [9]. The functional feature of these roots is the simplification of their anatomical structure, in particular the reduction of apical meristem and additional mechanical strength of the cell walls [8]. Apparently, the life of this cluster of suspended soils is limited both by the lifetime of epiphytes and in some cases and by the life of skeletal branches or the tree itself, i.e., many tens of years. Once on the ground, suspended soils undergo rapid degradation, as some experiments with the difference between cellulose degradation on the ground and in the crowns of a tropical forest have confirmed [24].

### 3.2. Elemental Composition

Data of the elemental analyses and composition of the organic materials and humic substances are presented in Table 3. In general, the humic acids contain more nitrogen and hydrogen but less oxygen and carbon. Although we are comparing sets of organic precursors with group of HAs, it is evident that the differences are significant for C (F = 8.95, $p < 0.0006$), H (F = 8.90, $p < 0.0006$), N (F = 292.70, $p < 0.0001$) and O (F = 7.50, $p < 0.002$). This may be interpreted as the result of the deep transformation of organic matter during humification. This is quite important for understanding the that fact that humification appears in suspended soils and could be considered as a universal process. In addition, characterising the structural and chemical composition of the suspended soils should be noted along with the content of trace elements and metal ions. Thus, the content of mobile $p$ varied from 267 to 691 mg/kg, mobile K 1091–4650 mg kg$^{-1}$, exchangeable Mg < 0.20–52.5 mmol 100 g$^{-1}$, exchangeable Ca 11.2–58. 8 mmo 100 g$^{-1}$, bulk Cu 1.62–17.6 mg kg$^{-1}$, Pb 1.59–11.5 mg kg$^{-1}$, Zn 12.8–89 mg kg$^{-1}$, Ni 0.97–54.2 mg kg$^{-1}$ and Cd < 0.005–0.105 mg kg$^{-1}$ (unpublished data). This suggests that the organic matter of suspended soils has strong buffer properties and can deposit ions of macronutrients, trace elements and heavy metals, which are most likely preserved in the form of complex complexes that apparently prevents their leaching. This is a fundamental point, as suspended soils are capable of retaining large masses of water due to their moisture capacity [5] and otherwise, given the vertical localization of suspended soils on the tree trunk, would be actively destroyed by water flows and washouts along the tree trunk.

**Table 3.** Elemental composition of humic acids and organic precursors.

| Sample Code | C, % | N, % | H, % | O, % | C/N | H/C | O/C |
|---|---|---|---|---|---|---|---|
| 1-1 | 51.02 * ± 0.05 | 0.84 ± 0.07 | 5.58 ± 0.21 | 42.56 ± 0.30 | 60.74 | 1.22 | 0.83 |
| 1-2 | 49.84 ± 0.30 | 0.98 ± 0.05 | 5.99 ± 0.01 | 43.18 ± 0.34 | 50.86 | 1.23 | 0.87 |
| 1-3 | 48.32 ± 0.58 | 0.51 ± 0.02 | 5.67 ± 0.14 | 45.49 ± 0.57 | 94.74 | 1.34 | 0.94 |
| 1-4 | 50.52 ± 0.19 | 0.58 ± 0.05 | 5.39 ± 0.15 | 43.51 ± 0.06 | 87.10 | 1.36 | 0.86 |
| 2-1 | 48.00 ± 0.10 | 0.74 ± 0.09 | 5.69 ± 0.03 | 45.55 ± 0.14 | 64.86 | 1.38 | 0.95 |
| 2-2 | 43.14 ± 0.57 | 1.78 ± 0.07 | 4.76 ± 0.05 | 50.32 ± 0.61 | 24.23 | 1.67 | 1.17 |
| 2-3 | 34.39 ± 1.27 | 1.53 ± 0.02 | 3.97 ± 0.06 | 60.10 ± 1.20 | 22.48 | 2.44 | 1.74 |
| 2-4 | 47.66 ± 1.10 | 1.39 ± 0.02 | 4.89 ± 0.07 | 46.04 ± 0.08 | 34.29 | 1.39 | 0.97 |
| 2-5 | 51.72 ± 0.17 | 0.83 ± 0.08 | 5.51 ± 0.03 | 41.92 ± 0.21 | 62.31 | 1.29 | 0.81 |
| 2-6 | 37.07 ± 0.56 | 1.58 ± 0.09 | 4.83 ± 0.06 | 56.51 ± 0.08 | 23.46 | 2.14 | 1.52 |
| 2-7 | 46.39 ± 0.92 | 1.38 ± 0.02 | 5.63 ± 0.19 | 46.59 ± 0.21 | 33.61 | 1.46 | 1.00 |
| HA 1-1 | 49.47 ± 0.58 | 3.91 ± 0.08 | 5.36 ± 0.17 | 41.26 ± 0.47 | 12.65 | 1.22 | 0.83 |
| HA 1-2 | 49.84 ± 0.28 | 3.59 ± 0.30 | 5.76 ± 0.12 | 40.80 ± 0.74 | 13.88 | 1.20 | 0.82 |
| HA 2-1 | 48.81 ± 0.24 | 3.99 ± 0.01 | 5.71 ± 0.05 | 41.48 ± 0.24 | 12.24 | 1.25 | 0.85 |
| HA 2-2 | 48.71 ± 0.13 | 4.01 ± 0.05 | 5.69 ± 0.07 | 41.59 ± 0.08 | 12.14 | 1.25 | 0.85 |

* Data on the elementary composition of humic acids are provided in percentages according to the recommendation of the International Humic Substances Society). (https://humic-substances.org/elemental-compositions-and-stable-isotopic-ratios-of-ihss-samples/, accessed on 1 March 2022). For recalculation to g kg$^{-1}$ in SI units, one can multiply percentages values by 10.

The representation of atomic ratios is given in Figure 5. It is evident from Figure 5 that the group of humic substances is more homogenous in comparison with the initial organic matter of both sample sets. At the same time, the initial organic matter is characterized by a wide range of atomic ratios, which is quite logical since humification leads to the selection of thermodynamically stable molecules; in the course of this process their elemental composition is homogenized. This well corresponds with theory of humificiation from D.S. Orlov [20] and analysis of data on humic substance content provided by [25–27].

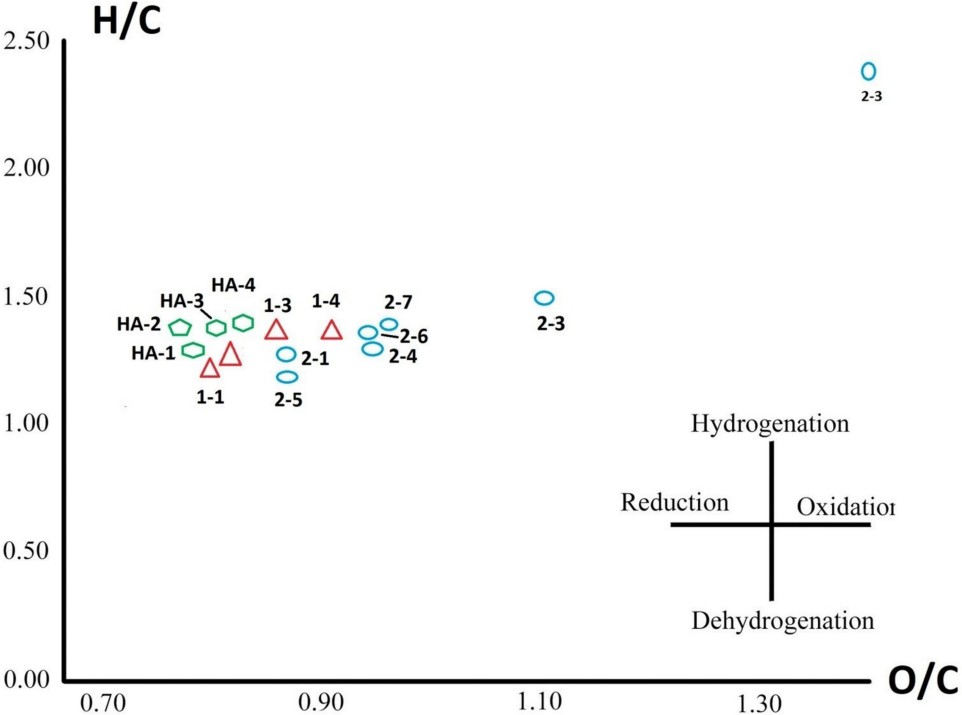

**Figure 5.** Van Krevelen diagram of the humic substances and organic materials.

Data on structural composition of organic matter and humic acids provided in Figure 6 and in Table 4. A set of structural fragments has been identified by CP/MAS 13C NMR spectroscopy: carboxyl groups (-COOR), carbonyl fragments (-C=O), CH3-, CH2-, CH- aliphatic carbon species, -C-OR of alcohols, esters and carbohydrates as well as phenolic carbon groups (Ar-OH), quinone (Ar=O) and aromatic (Ar-) groups, which demonstrates high heterogeneity and complexity of initial organic matters and of HAs isolated from various initial organic materials. The presence of the mentioned structural fragments in both initial organic materials and humic acids demonstrates the unity of the structural composition of all organic matter of epiphytic suspended soils.

Seven chemical groups in organic matter were identified by 13C-NMR spectroscopy in initial organic materials and HAs: nonpolar alkyl (0–46 ppm), N-alkyl/methoxyl (46–60 ppm), O-alkyl and anomer carbon compounds (60–110 ppm), C, H-substituted aromatic fragments (110–144 ppm), O, N -substituted aromatic fragments (144–160 ppm), carboxyl, esters, amides (160–185 ppm) and quinones (185–200 ppm) [18].

In general, the key difference of the HAs from initial organic materials is the increased portion of aromatic carbon species and a higher ratio of aryl compounds to aliphatic ones. This is the first attribute of the humification process [25,26], which appears not only in "normal" terrestrial mineral soils but also in suspended soils of tropical ecosystems. Data obtained confirms previously published data about initial humification in suspended soils [27]. The aromaticity degree varied from 31 to 42% in humic acids and from 7 to 25% in initial organic materials. This corresponds well with the classical theory of humification [21,28] and does not oppose the theory of "contentious nature of soil organic matter" [29]. Thus, continual transformation of soil organic matter and formation of the hu-

mic acids is the last point of this process. The portion of the sum of the aliphatic group was lower in humic acids (58–69%) than in initial organic materials (75–93%). This indicates that there is decreasing of the aliphatic part of organic matter under the process of suspended soil formation and humification. The quinone groups demonstrate essential decreasing in humic acids. The same may be fixed for O-alkyl and anomer carbon compounds. On the other hand, the group of nonpolar alkyl (0–46 ppm) increased in humic acids in comparison with the initial organic material. Moreover, the above-mentioned differences are evident from the shape of 13-C NMR spectra given in Figure 4. Here, there is a shift from the aliphatic maximum dominance in the case of bulk initial organic matter to the aromatic part in humic materials isolated from suspended soils. 13-C NMR spectra of suspended soils are comparable with those isolated from topsoils of Podzols of the western European plain [16] and Luvisols of the Privolzhskaya Upland [30]. In the case of a blind experiment, if the origin of humic acids is not known, they can be identified as humic substances isolated from podzolic soils (Podzols, Luvisols (Retisols). The content of C, H-substituted aromatic fragments was essentially higher than the group of O, N-substituted aromatic fragments in both humic acids and initial organic materials, which corresponds well with data on forest soils [17]. The structural composition of bulk organic matter derived from spectroscopic data is similar to very low humified materials from normal terrestrial soils of temperate or subarctic soils [31,32]. Thus, we can conclude, that suspended soils can serve as normal terrestrial soil in terms of litter fall material conversion. This conversion expressed in humification what may be considered as a universal mechanism for natural biotechnology—transformation and stabilization of organic matter in suspended soils of the tropical rain forest. Moreover, organic matter stabilization process take place not only in soil forest floor [33] but also on the various levels of epiphytic tropical vegetation communities. Suspended soils of the tropical forest may be considered as a very promising object of current so-called "humiomic science" [34]. In addition, this work provides an essential contribution to the estimation of carbon turnover rate and soil stocks in the concept of "biotic turnover" [35], in which the suspended parts were underestimated for many years. Additionally, in some sense, the study leads the science of humus beyond the limits of traditional soils because organic matter is accumulated and transformed not only in full-profile soils but also in soil biofilms [36] on the surface of glaciers [37], in waters [38] and, as our data showed, in suspended soils "separated" from the classical soil cover. The next step should be focused on investigation of 2-dimensional NMR spectra of humic acids and humification precursors, such as was performed for soils of unique ecosystems of tailgrass temperate rainforest in western Siberia [39].

**Table 4.** Carbon species in organic molecules according to the 13C NMR spectroscopy.

| Sample Code | 0–46, ppm | 46–60, ppm | 60–110, ppm | 110–144, ppm | 144–160, ppm | 160–185, ppm | 185–200, ppm | Al, * % | Ar, ** % | Ar/Al *** |
|---|---|---|---|---|---|---|---|---|---|---|
| 1-1 | 22 | 6 | 39 | 17 | 3 | 8 | 5 | 75 | 25 | 0.33 |
| 1-2 | 21 | 6 | 44 | 13 | 3 | 8 | 5 | 79 | 21 | 0.37 |
| 1-3 | 20 | 5 | 46 | 12 | 4 | 9 | 4 | 80 | 20 | 0.25 |
| 1-4 | 20 | 6 | 41 | 17 | 3 | 9 | 4 | 76 | 24 | 0.32 |
| 2-1 | 18 | 6 | 46 | 14 | 3 | 8 | 5 | 78 | 22 | 0.28 |
| 2-2 | 20 | 8 | 35 | 16 | 4 | 12 | 5 | 75 | 25 | 0.33 |
| 2-3 | 27 | 7 | 38 | 14 | 3 | 9 | 4 | 81 | 19 | 0.23 |
| 2-4 | 21 | 7 | 38 | 16 | 4 | 9 | 5 | 75 | 25 | 0.33 |
| 2-5 | 31 | 6 | 46 | 2 | 0 | 10 | 5 | 93 | 7 | 0.08 |
| 2-6 | 37 | 6 | 25 | 14 | 3 | 11 | 4 | 79 | 21 | 0.27 |
| 2-7 | 31 | 7 | 30 | 13 | 4 | 11 | 4 | 79 | 21 | 0.27 |
| HA1-1 | 30 | 6 | 18 | 31 | 5 | 9 | 1 | 63 | 37 | 0.59 |
| HA1-2 | 25 | 6 | 17 | 34 | 6 | 10 | 2 | 58 | 42 | 0.72 |
| HA2-1 | 28 | 9 | 21 | 25 | 5 | 10 | 2 | 68 | 32 | 0.47 |
| HA2-2 | 25 | 9 | 23 | 23 | 7 | 12 | 1 | 69 | 31 | 0.49 |

* Al—aliphatic carbon. ** Ar—aromatic carbon. *** Ar/Al—aromatic/aliphatic ratio.

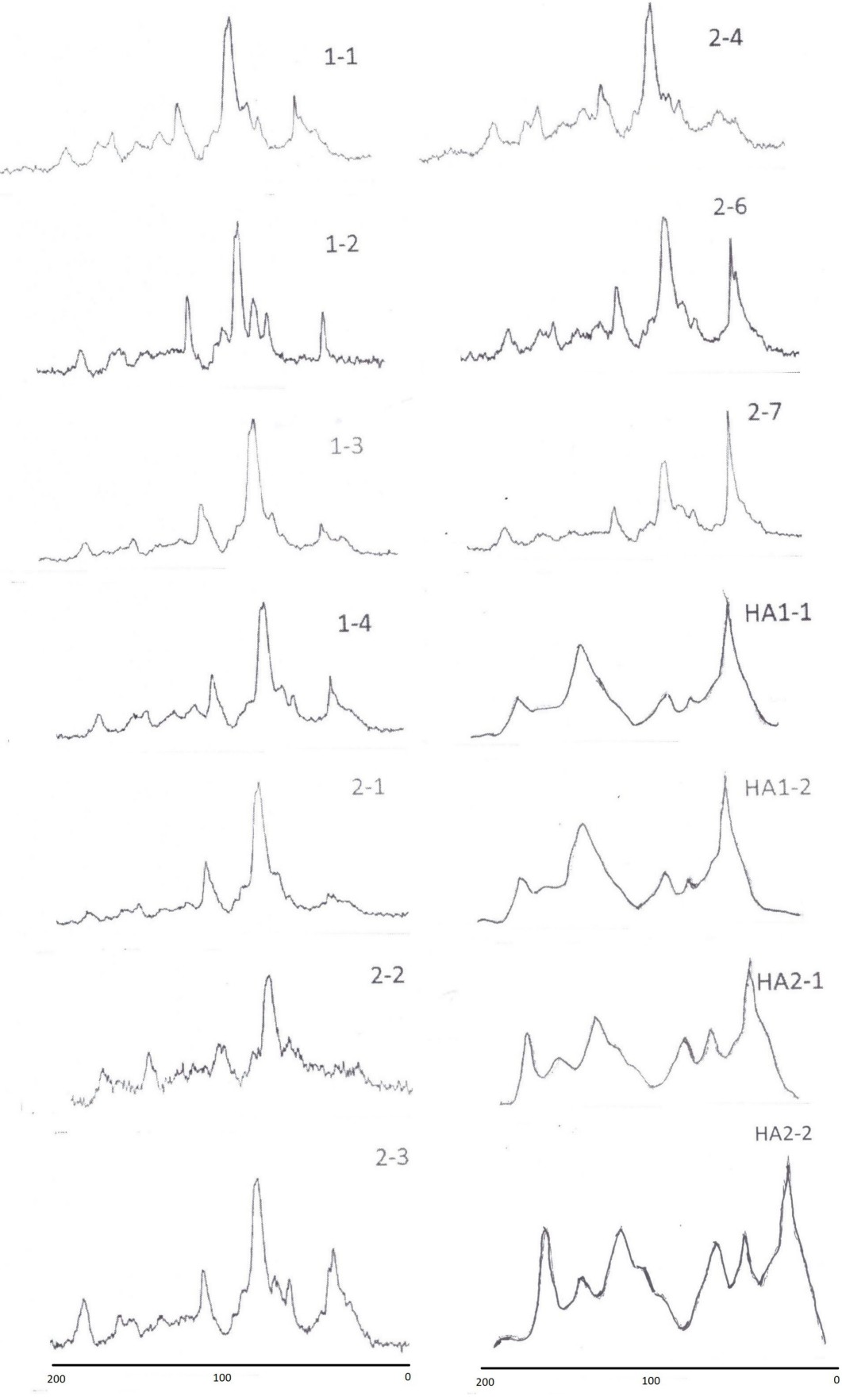

**Figure 6.** 13-C spectra of investigated organic materials (x-axis—chemical shift, ppm).

## 4. Conclusions

The organic matter of the planet exists in nature in various forms: standing biomass, soils, organic sediments and even in epiphytic layers of forest stands. The highest storages of biosphere organic matter are associated with classical terrestrial organo-mineral soils. Normally, the carbon of woody plant biomass is counted separately from soil as carbon of the standing biomass. At the same time, humification as a universal biogeochemical process already begins in plant residues still on the surface of the classical soil. Moreover, in tropical humid-forest ecosystems, most of the organic matter does not reach the soil surface at all and accumulates in the form of suspended soils. The data obtained in the present study characterize, for the first time, the component and structural composition of the organic matter of plant residues of suspended soils, as well as the products of their transformation—humic substances formed in suspended soils and derived from initial organic precursors. It has been shown that the aliphatic part dominates in the initial organic materials of suspended soils, whereas the humic substances are characterized by an increased fraction of aromatic fragments in the composition of molecules. Thus, even in the suspended soils, classical humification occurs, and this is not limited by the low content of mineral particles and cations in the suspended soils and the rather acidic reaction of the suspended pedoenvironment. Therefore, the existence of tiering and the formation of the corresponding layers of suspended soils is accompanied by stabilization and humification of organic matter that is accompanied by a radical change in its structural and component composition. There is stabilization of organic matter via the humifcation process, and this fact could be considered as a specific natural biotechnological process.

**Author Contributions:** E.A.—writing, experimental design, laboratory work, A.E.—field research, writing, editing. All authors have read and agreed to the published version of the manuscript.

**Funding:** The work was financial supported by the Russian Science Foundation (project #23-24-00037).

**Institutional Review Board Statement:** Not applicable.

**Informed Consent Statement:** Not applicable.

**Data Availability Statement:** Not applicable.

**Acknowledgments:** The work of E. Abakumov was partially supported by St. Petersburg State University (pr. No GZ_MDF_2023-1. ID PURE 101662710). Authors are grateful to Scientific Park of St. Petersburg State University, Research Center "Magnetic resonance research center" (NMR analysis), "Chemical analyses and materials research center" (elementary analysis) and "Observatory of Environmental Safety" (the use of polarization microscope Leica DM 750-P). The authors are grateful to V. Polyakov (Department of Applied Ecology, St. Petersburg State University) for his help with the laboratory work. We are grateful to the Joint Russian–Vietnamese Tropical Scientific and Technological Center for organization of field work.

**Conflicts of Interest:** The authors declare no conflict of interest.

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
