# Peer review of "Organic Matter Structural Composition of Vascular Epiphytic Suspended Soils of South Vietnam"

_applsci, doi:10.3390/app13074473_

Round 1

Reviewer 1 Report

line 219-- correct to "sample sets"

line 223 correct to " analysis"

line 227  Correct to “provided”

241 Correct to " ratio”

245 Correct to "humic acids”

247 Correct to "oppose”

248  Delete “ there is”

255 correct to “above mentioned”

260-262- re-write sentence

268 correct to " not only on the forest floor but also”

288 correct to " “and”

Table 4 – add a footnote to describe the measured parameters.

Author Response

Dear reviewer!

Thank you for all your comments, all comments has been taken into account, below you can see detailed replies.

With kindest regards,

Authors.

Report 1

Comments and Suggestions for Authors

line 219-- correct to "sample sets" - Corrected

line 223 correct to " analysis" - Corrected

line 227  Correct to “provided” - Corrected

241 Correct to " ratio” - Corrected

245 Correct to "humic acids” - Corrected

247 Correct to "oppose” - Corrected

248  Delete “ there is” - Corrected

255 correct to “above mentioned” - Corrected

260-262- re-write sentence - Rewritten

268 correct to " not only on the forest floor but also” - Corrected

288 correct to " “and” - Corrected

 Table 4 – add a footnote to describe the measured parameters - Added

Reviewer 2 Report

1.     Please describe more significance of this study in the introduction of this manuscript.

2.     Some methods are not clear.

3.     Please add some explanation of your important results to the Discussion, in which the line 140 – 270 are not clear.

4.     Some sentences in this manuscript are not rigorous. Please revise English grammar in this manuscript.

5.     My specific comments are in the manuscript, please check them.

Author Response

Dear reviewer!

Thank you for all your comments, all comments has been taken into account, below you can see detailed replies.

With kindest regards,

Authors.

Comments and Suggestions for Authors

  1. Please describe more significance of this study in the introduction of this manuscript – information has been added
  2. Some methods are not clear.- has been clarified
  3. Please add some explanation of your important results to the Discussion, in which the line 140 – 270 are not clear - corrected
  4. Some sentences in this manuscript are not rigorous. Please revise English grammar in this manuscript- Done
  5. My specific comments are in the manuscript, please check them.

all comments from pdf file has been taken into account

Reviewer 3 Report

Dear Authors,

Despite all advantages of the research performed, the manuscript text contains several important shortcomings, and they should be eliminated.

A huge number of typos mistakes in the manuscript. Please, check all the text very attentively and carefully.

Some corrections have to be made in the lines: 37, 52, 66 (cculky na nac ? What it is ?), 126, 129, 155, 159, 160, 166, 168… title of figure 5, etc.

Section 2.2: the phrase “we collected” was used too frequently. Please replace ‘to collect’ by synonyms (to pick, gather…).

Table 2, 3 and 4: some obtained values have to be expressed with dots (not comas).

Citations of references 20 and 21 were missed in the text.

You are claiming that the aim of this work was to characterize structural and chemical composition of suspended soils of South Vietnam. For this reason it will be very important to know conductivity (µS/cm) of the soils and main inorganic composition (amounts of Ca, Mg, K, Na, Al, Mo, Mn, Cu, Fe, Zn, Ni, P.. and concentrations of some hazardous and heavy metals, such as Cd, Cr, Pb) in the investigated soil material. These parameters are very important for characterisation of the soils.

Please, check a reference list attentively.

Author Response

Dear reviewer!

Thank you for all your comments, all comments has been taken into account, below you can see detailed replies.

With kindest regards,

Authors.

Despite all advantages of the research performed, the manuscript text contains several important shortcomings, and they should be eliminated.

A huge number of typos mistakes in the manuscript. Please, check all the text very attentively and carefully. – thank you very much indeed! We have improved all mistapes and mistakes.

Some corrections have to be made in the lines: 37, 52, 66 (cculky na nac ? What it is ?), 126, 129, 155, 159, 160, 166, 168… title of figure 5, etc. - corrected

Section 2.2: the phrase “we collected” was used too frequently. Please replace ‘to collect’ by synonyms (to pick, gather…) – thank you, corrected.

Table 2, 3 and 4: some obtained values have to be expressed with dots (not comas) - corrected

Citations of references 20 and 21 were missed in the text. - corrected

You are claiming that the aim of this work was to characterize structural and chemical composition of suspended soils of South Vietnam. For this reason it will be very important to know conductivity (µS/cm) of the soils and main inorganic composition (amounts of Ca, Mg, K, Na, Al, Mo, Mn, Cu, Fe, Zn, Ni, P.. and concentrations of some hazardous and heavy metals, such as Cd, Cr, Pb) in the investigated soil material. These parameters are very important for characterisation of the soils. – Information has been added

Please, check a reference list attentively. – Corrected, but we were not able to measure resistivity on the height due to technical reasons, soil chemistry data has been amended

Reviewer 4 Report

Dear authors,

The topic is interested and ecologically is high important specially in specific ecosystems. However, there are some comments that should be considered before publishing such as:

The abstract should be rewritten because it seems like an abstract of review not research paper

There is a missed full stop in line 37

Please convert the units into SI unit such as Mg ha-1 and kg ha-1

More details are needed in the material and methods specially for the equipments and the methods used to determine N, C, O and H

In table 3, please convert the % into g kg-1 and put the significance letters in the table

Deep discussion is needed specially in section 3.1 and the first part of section 3.2

The paper should be supported by recent references from 2020 till now

Author Response

Dear reviewer!

Thank you for all your comments, all comments has been taken into account, below you can see detailed replies.

With kindest regards,

Authors.

The topic is interested and ecologically is high important specially in specific ecosystems. However, there are some comments that should be considered before publishing such as:

The abstract should be rewritten because it seems like an abstract of review not research paper - - Rewritten

There is a missed full stop in line 37 – corrected.

Please convert the units into SI unit such as Mg ha-1 and kg ha-1 - corrected

More details are needed in the material and methods specially for the equipments and the methods used to determine N, C, O and H – has been detalized.

In table 3, please convert the % into g kg-1 and put the significance letters in the table – here we are disagree, g kg-1 is typical for bulk samples of soil, but according to recommendation of International Humic Substances the elementary composition of humic substances normally is presented in percentages (e.g. https://humic-substances.org/elemental-compositions-and-stable-isotopic-ratios-of-ihss-samples/)

Deep discussion is needed specially in section 3.1 and the first part of section - Done

The paper should be supported by recent references from 2020 till – few references has been added.

Round 2

Reviewer 3 Report

Dear Authors,

English language and style still required minor corrections.

Author Response

Dear reviwer! 

thank you, english has been improved

Dear Authors,

English language and style still required minor corrections.

Reviewer 4 Report

There are two comments still need to be responded: changing % in table 3 to units such as g kg-1 as well changing my/kg to mg kg-1 through the manuscript

Author Response

Dear reviewer, we agree that you usually need to follow the SI system of units, but in this case we followed the recommendations of the International Humic Substances Society  in order to make the results comparable

Thank you, you comment is very usefull if one will recalculate to balance of humic acids stock in whole ecosystems

we have added comment to 

table footnote:

* Data on elementary composition of humic acids are provided in percentages according to recommendation of International humic substances society). (https://humic-substances.org/elemental-compositions-and-stable-isotopic-ratios-of-ihss-samples/). For recalculation to g kg-1 in SI units one can multiply percentages values by 10.